# Long-Term Accuracy Enhancement of Binary Neural Networks Based on Optimized Three-Dimensional Memristor Array

**DOI:** 10.3390/mi13020308

**Published:** 2022-02-17

**Authors:** Jie Yu, Woyu Zhang, Danian Dong, Wenxuan Sun, Jinru Lai, Xu Zheng, Tiancheng Gong, Yi Li, Dashan Shang, Guozhong Xing, Xiaoxin Xu

**Affiliations:** 1Key Laboratory of Microelectronics Device & Integrated Technology, Institute of Microelectronics of Chinese Academy of Sciences, Beijing 100029, China; yujie@ime.ac.cn (J.Y.); zhangwoyu@ime.ac.cn (W.Z.); dongdanian@ime.ac.cn (D.D.); sunwenxuan@ime.ac.cn (W.S.); laijinru@ime.ac.cn (J.L.); zhengxu2018@ime.ac.cn (X.Z.); gongtiancheng@ime.ac.cn (T.G.); liyi@ime.ac.cn (Y.L.); shangdashan@ime.ac.cn (D.S.); gzxing@ime.ac.cn (G.X.); 2School of Microelectronics, University of Chinese Academy of Sciences, Beijing 100049, China

**Keywords:** embedded neural network, 3D vertical architecture, long-term accuracy, device engineering

## Abstract

In embedded neuromorphic Internet of Things (IoT) systems, it is critical to improve the efficiency of neural network (NN) edge devices in inferring a pretrained NN. Meanwhile, in the paradigm of edge computing, device integration, data retention characteristics and power consumption are particularly important. In this paper, the self-selected device (SSD), which is the base cell for building the densest three-dimensional (3D) architecture, is used to store non-volatile weights in binary neural networks (BNN) for embedded NN applications. Considering that the prevailing issues in written data retention on the device can affect the energy efficiency of the system’s operation, the data loss mechanism of the self-selected cell is elucidated. On this basis, we introduce an optimized method to retain oxygen ions and prevent their diffusion toward the switching layer by introducing a titanium interfacial layer. By using this optimization, the recombination probability of Vo and oxygen ions is reduced, effectively improving the retention characteristics of the device. The optimization effect is verified using a simulation after mapping the BNN weights to the 3D VRRAM array constructed by the SSD before and after optimization. The simulation results showed that the long-term recognition accuracy (greater than 10^5^ s) of the pre-trained BNN was improved by 24% and that the energy consumption of the system during training can be reduced 25,000-fold while ensuring the same accuracy. This work provides high storage density and a non-volatile solution to meet the low power consumption and miniaturization requirements of embedded neuromorphic applications.

## 1. Introduction

Energy is a crucial resource for smart devices in the Internet of Things (IoT), as most applications are powered by batteries or use energy-harvesting techniques [1,2,3,4]. Because of this, energy-efficient artificial intelligence technologies are becoming increasingly important for the IoT. Since Deep Neural Networks (DNNs) require a high bandwidth, large memory capacity, and large power consumption, running DNNs on target embedded systems and mobile devices has become a challenge [5,6,7,8,9,10]. In comparison, Binarized Neural Networks (BNN) can significantly reduce computational complexity and memory consumption while having satisfactory accuracy on various image datasets [11]. In embedded IoT systems, neural networks must be able to perform pre-trained cognitive tasks in an efficient way. In this case, the weights of the trained neural network should remain unchanged and only limited in-field updates should be performed. Currently, the resource consumption by add-ons has become a limitation in memristor-based analog computing in memory systems. Analog designs require additional circuits, such as analog-to-digital and digital-to-analog converters, to fight against undesirable device properties. In contrast, binary networks offer obvious advantages in terms of speed, energy consumption, memory occupation and other aspects. However, more storage units are required as storage for the weights in BNNs. Although several software algorithms, such as sparse mapping schemes, have been proposed to address the large number of weights [12,13], neuromorphic architectures still demand a high amount of storage. Against this background, a 3D memristor array would be the most effective hardware scheme for maximizing the area’s efficiency [14,15,16]. However, the retention of written data on the device could affect the energy efficiency of the system’s operation [15,16].

The 3D architecture for a memristor includes both stacked and vertical arrays. The latter, also known as vertical RRAMs (VRRAMs), are patterned with fewer lithography steps and therefore less costly. The self-selected device (SSD) is the only way to overcome sneak current in 3D vertical integration. The currently reported SSDs, which are based on interface barrier modulation, possess good uniformity but poor retention performance [17,18,19,20]. Retention failures have an adverse impact on the overall accuracy of the implemented neural network, which worsens when the device operates at high temperatures during edge computing [11]. The undesirable properties of devices have also hindered the practical application of emerging memristors. In order to restore the recognition accuracy, the network needs to be retrained when the accuracy drops to a certain level. The additional power consumption of the refresh process is in direct contrast to the low power requirements of edge computing. Therefore, the retention issue in 3D memristor arrays is highly non-trivial, especially for long-term accuracy, and needs to be considered and addressed.

In this work, firstly, we fabricated SSDs with the TiN/TiO_x_/HfO_x_/Ru and TiN/TiO_x_/HfO_x_/Ti/Ru structures, and then tested the electrical properties including operation voltages, nonlinearity and retention characteristics. We proposed that the underlying mechanisms of LCSs (low-conductance states) and HCSs (high-conductance states) are due to Poole–Frenkel (PF) emission and trap-assisted tunneling (TAT). Then, we brought the mechanism of retention degradation forward through a series of comparative experiments. By introducing a deep-level trap in this structure with a Ti interfacial layer, the oxygen ions are firmly trapped, and retention is highly developed. Thirdly, we simulated the long-term accuracy of the BNN through fashion-MNIST tasks by mapping binarized weights to the non-volatile 3D memristor arrays constructed by these SSDs. Due to the improved retention characteristics of the device, the trained network can guarantee a good recognition accuracy in 10^5^ s, which is estimated as up to 3 years. At the same time, this scheme can reduce the energy consumption of training the network by a factor of 25,000, since the network needs to be retrained when the recognition accuracy falls below 80%. This optimized 3D vertical RRAM offers an option to provide high storage density and a non-volatile means to meet the requirements of embedded neuromorphic applications for low power consumption and miniaturization.

## 2. Experiment

The 3D vertical memristor array was fabricated following the process described in previous work [19]. Firstly, multiple TiN (60 nm)/SiO_2_ (100 nm) layers were deposited by PVD and PECVD. Then, the trenches with smooth sidewall profiles were formed by one-step etching. TiN layers serving as the bottom electrode of the self-selected cell were denoted as word lines (WLs) in the 3D memory array. The selective layers were prepared by a plasma oxidation process and self-aligned with the bottom electrode, and their thicknesses were controlled by the adjustment of the oxidation time. After depositing the HfO_2_ and switching layers with different thicknesses on the sidewalls using atomic layer deposition (ALD), the top electrode W (60 nm) with a line width of 1 μm was sequentially deposited by magnetron sputtering. Finally, stair-like WLs were opened by selectively etching in sequence, and bit-lines (BLs) were formed using a lift-off process. The area of the memory cell was defined by the thickness of the bottom electrode (TiN) and the width of the lateral BL—i.e., 0.06 um^2^. Figure 1a shows a schematic diagram of an inference accelerator for the embedded neuromorphic application based on the 3D vertical memristor array. By implementing a BNN with 3D VRRAM, the neural networks executed a variety of recognition tasks with high density and low power consumption. Figure 1b shows a 3-layer fully connected network based on a BNN for Fashion-MNIST classification tasks. The hidden layer consisting of 64 × 128 neurons was implemented by 3D VRRAM. The detailed structure of the SSD was composed of TiO_x_ as the selective layer (SL) and HfO_x_ as the memory layer (ML). To investigate the failure mechanism of the device, TiO_x_ layers with different thicknesses of 0 nm, 10 nm and 30 nm were designed by adjusting the oxidation time, and HfO_x_ layers with thicknesses of 2 nm, 3 nm and 5 nm were fabricated. Four-layer VRRAM arrays were fabricated with the dimensions of 4 × 8 × 32. To gain insight into the internal structure of the device, the chemical composition of the material stack was analyzed using the energy dispersive X-ray (EDX) spectrum obtained by line scanning from TiN electrodes to Pd electrodes, as shown in Figure 1c. The curve in the inset shows that the peak of Ti was observed at ~55 nm, indicating that the Ti layer was inserted between the switching layer and the top electrode.

## 3. Results

Electrical tests were performed in the previously fabricated 3D VRRAM arrays. Figure 1d shows the I–V curves of the device with the Ti interfacial layer (DWT) and without the Ti interfacial layer (DOT). Both devices exhibit typical bipolar characteristics. The parameters are clearly defined in Figure 1d. The DOT switched from an LCS to an HCS when a voltage of +6V was applied to the TE. However, this programming voltage was decreased to +5V for the DWT when an interfacial layer was used. To ensure that the resistance state of the device was not affected, the read voltages for the DOT and the DWT were chosen to be 2 V and 1.5 V, respectively. The on/off ratios collected from 20 different devices are shown in Figure 1e. It can be seen that the on/off ratio for the DWT was 1 order higher than that of the DOT, primarily due to the increased resistance of the low-resistance state, which is consistent with the I–V characteristics in Figure 1d. The nonlinear ratio (NR) is defined as the ratio of the currents at V_read_ and 1/2V_read_. As shown in Figure 1e, the NR for the DOT was calculated from the currents at 2 V and 1 V, while for DWT, the NR was calculated at 1.5 V and 0.75 V. The nonlinearity of the DWT device was smaller than that of the DOT device. Therefore, the integration scale will be sacrificed when the SSD is integrated into a 3D architecture.

Figure 2a shows the switching mechanisms for these devices. For LCSs at a positive voltage, the electrons injected from TiN will be trapped at a low positive voltage. The competition between trap and de-trap leads to an ultra-low leakage current. The Poole–Frenkel (PF) emission [21] dominates in this phase and the current density *J_PF_* can be expressed by the following formula:(1)$$J_{PF}=q\mu N_{C}E\exp[\frac{-q(\phi T-\sqrt{qE/\pi \varepsilon })}{kT}]$$
where *μ* represents the electronic drift mobility, *Nc* represents the density of states in the conduction band, *E* represents the applied electric field, *ΦT* represents the depth of the trap’s potential well, *k* represents Boltzmann’s constant and *T* represents the absolute temperature. As the voltage increases, more oxygen vacancies are trapped in the switching layer, serving as the transition level. At this point, trap-assisted-tunneling (TAT) [21] controls the conduction and the device switches to an HCS. TAT can be expressed by the formula:(2)$$J_{TAT}=A\exp(\frac{-8\pi \sqrt{2qm^{*}})}{3hE}\phi_{T}^{3/2})$$
where *A* refers to a constant value and *Φ_T_* represents the energy of the electron traps with respect to the oxide’s conduction band edge. When a negative bias is applied to the device, the electron injection from the Pd electrode is suppressed by a Schottky barrier. The trapped electrons can be released into the conductance band through tunneling at high electric fields and the device switches back to a LCS. Both LCSs and HCSs at negative voltages follow Schottky emission, leading to the device’s high rectification ratio.

To elucidate the mechanism of data loss in the DOT, the devices with varied SL/ML stacks were investigated. Figure 2c,d show the I–V curves of these devices with different structures. The current of the devices decreases sharply as the thickness of HfO_x_ (or TiO_x_) increases. When the thickness of HfO_x_ is 2 nm, the curve of the LCS almost overlaps with that of the HCS. In this case, the electrons trapped in the HfO_x_ layer can easily tunnel through the HfO_x_ layer in a short time, resulting in no significant effect on the memory. In the case where TiO_x_ is 0 nm, the trapped electrons can easily run away by tunneling through both sides, leaving an obvious no-memory behavior. This result is consistent with the data shown in Figure 2c. Therefore, the presence of TiO_x_ can help to suppress the loss of trapped electrons.

## 4. Discussion

As discussed above, the schematic diagram of the retention loss of the DOT is demonstrated in Figure 2b. During the measurement of retention, electrons trapped at sites near the interface are released through tunneling, while the electrons in the middle are emptied by thermal emission, resulting in data degradation in the HCS. According to this mechanism, the retention characteristics are closely related to the energy level of the trap sites, which can be enhanced reasonably by introducing deep-level traps. The Ti layer in the proposed device is an effective approach to implement a deep-level trap [22]. Figure 3a demonstrates the retention test results on the DWT and the DOT, where the DWT showed almost no degradation within 5 × 10^4^ s, much better than that of the DOT (2000 s). Titanium has a strong ability to absorb oxygen and introduce oxygen vacancies. During the retention test for the DWT, the titanium interfacial layer retained oxygen ions and prevented their diffusion toward the switching layer. In this case, the recombination probability of Vo and oxygen ions was reduced, effectively improving the retention characteristics of the HCS of the SSD. In addition, titanium is a CMOS-compatible material and can be widely used in mass production. Therefore, a Ti interface layer insertion is a practical and effective optimization method for 3D VRRAM arrays.

The statistical results of the retention testing for the DWT and the DOT are shown in Figure 3b,c. For the DWT, the distributions of the LCS and the HCS were fully overlapped after 1000 s at room temperature (RT). In contrast, the retention evaluation of the DWT was performed at a high temperature (85 °C). Even after 30 h of baking, there was still a 10-fold difference in conductance between the LCS and the HCS. To evaluate the lifetime of the DWT, three high temperatures were adopted to accelerate aging. By extrapolating the Arrhenius plot, the lifetime of the DWT at RT was calculated to be 3 years, as shown in Figure 3d, indicating that the introduction of the Ti layer greatly improves data retention properties.

To evaluate the effectiveness of this device structure optimization on the long-term accuracy of the trained network, a multi-layer perceptron BNN was implemented for the Fashion-MNIST classification task using a 3D VRRAM array, as shown in Figure 2b. The weights of the BNN were constrained to +1 or −1 by the activation function and were then mapped to the 3D VRRAM array. When the conductance of the device decayed over time to a pre-set threshold, the weights stored in the device became invalid. During inference, the fading weights reduced the systematic accuracy of the trained BNN. Figure 4a shows that the recognition rate of the BNN decreased with inference time. For the network implemented by the DOT, the recognition rate dropped rapidly to below 60% after 10^4^ s. For the DWT, the recognition rate remained at 84% when the inference time exceeded 10^5^ s, achieving an improvement of 24%. According to the DWT lifetime evaluated in Figure 3d, the BNN implemented with a DWT can maintain a satisfactory accuracy of more than 80% over 3 years without any refresh operations, thus achieving an ultra-low power consumption. In this scenario, we propose a hypothesis that the network needs to be retrained when the recognition accuracy drops below 80%. According to this premise, we calculated the comparison of the training consumption for the BNN implemented with the DWT and the DOT. Figure 4b shows the energy consumed by BNN inference in Fashion-MNIST classification tasks. The HCS of the DWT was larger than that of the DOT, resulting in more energy being needed for a one-step identification operation. However, the BNN network constructed by the DOT needed to be constantly refreshed due to the poor retention characteristics of the DOT. A refresh operation corresponds to the programming operation of the device, which consumes much more power than a read operation. For picture inference, the DWT saved 25,000 times more energy than the DOT. Therefore, the high-density 3D VRRAM constructed by DWT is suitable for power-sensitive edge computing applications.

## 5. Conclusions

Due to its high-density storage and non-volatile properties, 3D VRRAM is suitable for neural network hardware implementation in embedded neuromorphic IoT systems. In order to improve the long-term accuracy and reduce the refresh rate of BNNs after training, an optimized 3D VRRAM with a TiN/TiO_x_/HfO_x_/Ti/Ru structure is proposed to implement a BNN. Since the failure of the DOT is a process of electron de-trapping near the interface, the retention of the DWT is improved by introducing a deep-level trap through the Ti interface layer. By using this DWT-based 3D RRAM, BNNs can be constructed to represent long-term inference with a high accuracy and ultra-low training power consumption. This work provides a solution for high-density neural network implementation for edge computing.

## Figures and Tables

**Figure 1 micromachines-13-00308-f001:**
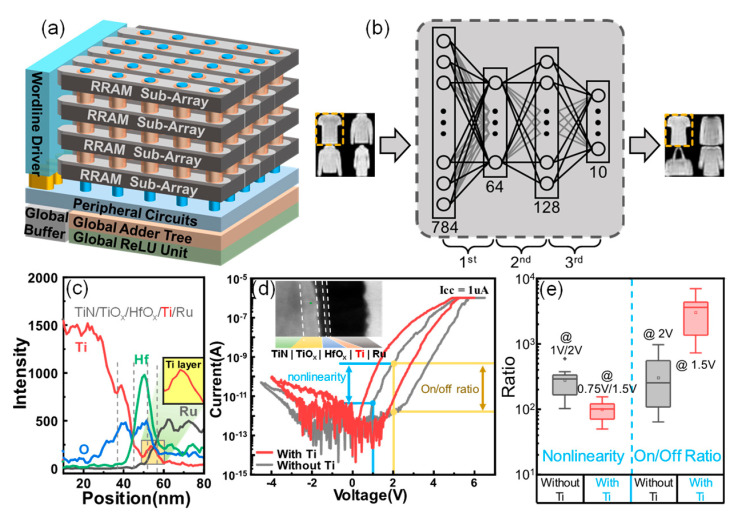
(**a**) A system-level design of the NN accelerator. The weights of each layer in the neural network are mapped to different sub-arrays of the 3D memristor array. (**b**) Architecture diagram for evaluating the neural network with Fashion-MNIST classification tasks. (**c**) EDX line scan on the cross-section of the device with the structure of TiN/TiOx/HfOx/Ti/Ru. The insert shows a scaled version of the Ti signal peak. (**d**) Typical I–V curves of device with Ti interlayer (DWT) and device without Ti interlayer (DOT). The insert shows a TEM image of the DWT. (**e**) The distribution of nonlinearity and on/off ratio for the DWT and the DOT.

**Figure 2 micromachines-13-00308-f002:**
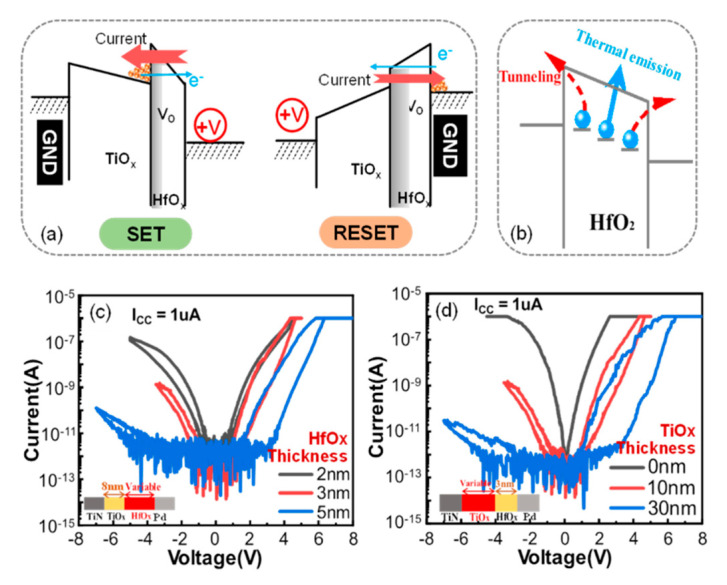
(**a**) Schematic diagram of the set/reset switching mechanism. (**b**) Schematic diagram of the retention loss of the self-selected device. (**c**) I–V curves of devices with different HfO_x_ thicknesses, with TiO_x_ fixed at 8 nm. (**d**) I–V curves of cells with different TiO_x_ thicknesses, with the HfO_x_ layer fixed at 3 nm.

**Figure 3 micromachines-13-00308-f003:**
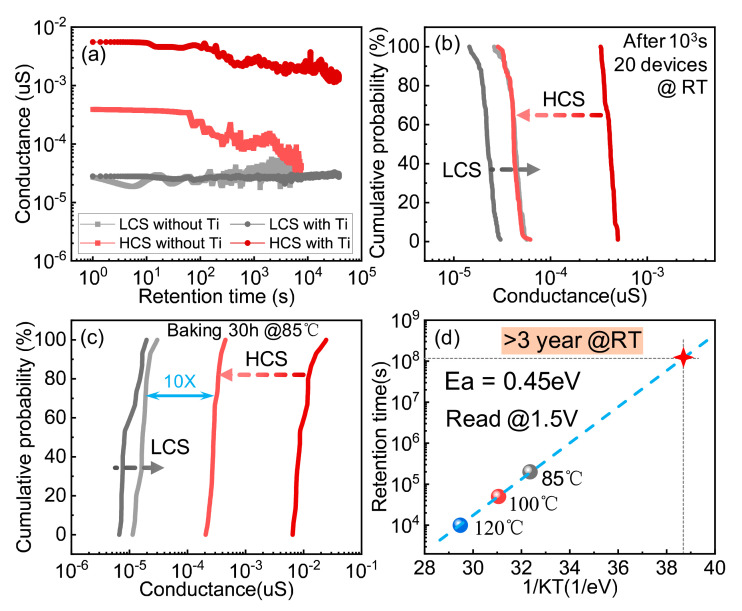
(**a**) Retention tests of the DWT and the DOT IL layers. (**b**) The distribution of the HRS and the LRS of the DOT before and after 1000 s at room temperature. (**c**) The distribution of the HRS and the LRS of the DWT before and after baking at 85 °C for 30 h. (**d**) Arrhenius plot based on the DWT’s lifetime at high temperature. The device can retain stored data for more than 3 years at room temperature.

**Figure 4 micromachines-13-00308-f004:**
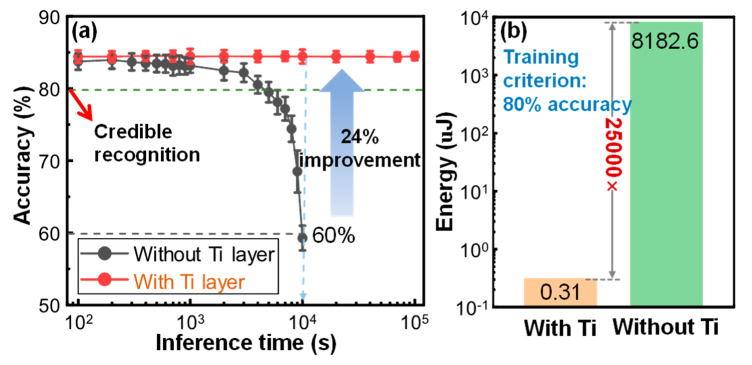
(**a**) The recognition rate decreased with inference time after mapping the weights of the BNN to the 3D VRRAM array constructed with the DWT and the DOT at room temperature. (**b**) According to this premise, the energy consumed by BNN implemented with the DWT and the DOT for Fashion-MNIST classification tasks required retraining when the recognition accuracy dropped below 80%.

## Data Availability

The data that support the findings of this study are available from the corresponding author upon request.

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
