# Peer review of "Long-Term Accuracy Enhancement of Binary Neural Networks Based on Optimized Three-Dimensional Memristor Array"

_micromachines, 2022, doi:10.3390/mi13020308_

Round 1

Reviewer 1 Report

The authors present a memristor solution with improved long-term accuracy and a significantly reduced power consumption. The retention issue is largely mitigated by interfacing the additional Ti layer in the architecture. Moreover, a systematic study of device structure-performance correlation reveals the mechanism of the data loss, and the data fully supports the proposed claim. However, a few scientific aspects are either missing or not clarified with details in the manuscript. The work can be considered to get published after addressing the below issues.

  1. It would be beneficial to the readers if the authors could briefly review the work/methods developed to address the retention issue in the field. Can the authors also compare the figure of merit between previous work and this work?
  2. Can the authors explain the rationale for choosing Ti over other materials to create a deep trap state? It is interesting to know if other metals can do better than Ti.
  3. A minor editorial suggestion: The caption descriptions of Figure 2c and 2d do not match with the contents and need to be swapped.

Author Response

Revisions to Manuscript ID micromachines-1581488

We appreciate your efforts in the timely review of our paper. We'd like to resubmit our revision of Manuscript ID micromachines-1581488, entitled " Enhancement of long-term accuracy for binary neural networks based on optimized three-dimensional memristor array”, including revisions to the concerns brought up by the reviewers. The editor’s and reviewers’ suggestions were very helpful and served to improve the manuscript. According to their suggestions and feedback, we have completed a major revision to our manuscript. The detailed answers to all the comments are summarized as follows:

Comments from Reviewer 1

The authors present a memristor solution with improved long-term accuracy and a significantly reduced power consumption. The retention issue is largely mitigated by interfacing the additional Ti layer in the architecture. Moreover, a systematic study of device structure-performance correlation reveals the mechanism of the data loss, and the data fully supports the proposed claim. However, a few scientific aspects are either missing or not clarified with details in the manuscript. The work can be considered to get published after addressing the below issues.

Comment 1: It would be beneficial to the readers if the authors could briefly review the work / methods developed to address the retention issue in the field. Can the authors also compare the figure of merit between previous work and this work?

Reply to Comment 1: Thank you very much for your valuable comment.  There are various approaches to address the retention issues in the field. We have investigated previous work and conclude the table below.

Approaches to address the retention issue
ref Device Method merit
[1] TiN/TaOx/HfO2/TiN Processing technology: annealing in the Oxygen atmosphere Simple but uncontrollable process
[2] TiN/TaO/AlOx/TiN Interfacial engineering: introduction of AlOx barrier layer Nonstandard CMOS materials
[3] Ir/Ta2O5-δ/TaOx/TaN Operation algorithm optimization: 2-step-forming Increased circuit consumption
[4] TiN\Ta2O5\Ta Processing technology: Annealing in NH3 atmosphere Non-standard annealing atmosphere
[5] HfO2 based RRAM Operation algorithm optimization: Pulse-width voltage-current write-verify-write (PVC-WVW) Increased circuit consumption
[6] HfO2 based RRAM Operation algorithm optimization: Low-voltage write-current-limiting-scheme Increased circuit consumption
This work Self-selective cell Interfacial engineering: introduction of Ti barrier layer Standard CMOS materials

 [1] X. Huang, H. Wu, D. C. Sekar, S. N. Nguyen, K. Wang and H. Qian, "Optimization of TiN/TaOx/HfO2/TiN RRAM Arrays for Improved Switching and Data Retention," 2015 IEEE International Memory Workshop (IMW), 2015, pp. 1-4, doi: 10.1109/IMW.2015.7150300[2] Lin, YD., Chen, PS., Lee, HY. et al. Retention Model of TaO/HfO x and TaO/AlOx RRAM with Self-Rectifying Switch Characteristics. Nanoscale Res Lett 12, 407 (2017). doi: 10.1186/s11671-017-2179-5[3] T. Ninomiya, Z. Wei, S. Muraoka, R. Yasuhara, K. Katayama and T. Takagi, "Conductive Filament Scaling of TaOx Bipolar ReRAM for Improving Data Retention Under Low Operation Current," in IEEE Transactions on Electron Devices, vol. 60, no. 4, pp. 1384-1389, April 2013, doi: 10.1109/TED.2013.2248157[4] L. Goux et al., "Role of the Ta scavenger electrode in the excellent switching control and reliability of a scalable low-current operated TiN\Ta2O5\Ta RRAM device," 2014 Symposium on VLSI Technology (VLSI-Technology): Digest of Technical Papers, 2014, pp. 1-2, doi: 10.1109/VLSIT.2014.6894401.[5] P. Jain et al., "13.2 A 3.6Mb 10.1Mb/mm2 Embedded Non-Volatile ReRAM Macro in 22nm FinFET Technology with Adaptive Forming/Set/Reset Schemes Yielding Down to 0.5V with Sensing Time of 5ns at 0.7V," 2019 IEEE International Solid- State Circuits Conference - (ISSCC), 2019, pp. 212-214, doi: 10.1109/ISSCC.2019.8662393.[6] C. -C. Chou et al., "An N40 256K×44 embedded RRAM macro with SL-precharge SA and low-voltage current limiter to improve read and write performance," 2018 IEEE International Solid - State Circuits Conference - (ISSCC), 2018, pp. 478-480, doi: 10.1109/ISSCC.2018.8310392.

Comment 2: Can the authors explain the rationale for choosing Ti over other materials to create a deep trap state? It is interesting to know if other metals can do better than Ti.

Reply to Comment 2: Thank you very much for your valuable comment. In the self-selective device, the high conductive state is dominated by the tunneling barrier which modulated by the oxygen vacancies. The degradation of HCS is due to the widening of the barrier caused by the recombination of oxygen ions and oxygen vacancies in the functional layer. Ti has strong oxygen absorption capacity beyond other metal material such as Hf, Pt, W or Ta. During the retention test, titanium retains oxygen ions and prevents diffusion of oxygen ions to the switching layer. In this case, the recombination probability of Vo and oxygen ions is reduced, which effectively improves the retention characteristics of the device. Additionally, the titanium is the CMOS-compatible material which could be widely used in the mass manufacturing. We have added the comment in the Discussion section of the revised edition. “During the retention test for DWT, titanium interfacial layer retains oxygen ions and pre-vents the diffusion of oxygen ions toward to the switching layer. In this case, the recombination probability of Vo and oxygen ions is reduced, effectively improving the retention characteristics of the HCS of the SSD. In addition, titanium is a CMOS-compatible material and can be widely used in mass production.”

[1] Y. Y. Chen, M. Komura, R. Degraeve, B. Govoreanu, L. Goux, A. Fantini, N. Raghavan, S. Clima, L. Q. Zhang, A. Belmonte, A. Redolfi, G. S. Kar, G. Groeseneken, D. J. Wouters and M. Jurczak, “Improvement of Data Retention in HfO2/Hf 1T1R RRAM cell under low operating current,”, in Proc. IEDM Tech. Dig., 2013, pp. 252-255.

[2] “Unveiling the Switching Mechanism of a TaOx/HfO2 Self-Selective Cell by Probing the Trap Profiles with RTN Measurements”

Comment 3: A minor editorial suggestion: The caption descriptions of Figure 2c and 2d do not match with the contents and need to be swapped.

Reply to Comment 3: Thank you very much for your kindly reminder. We have swapped the positions of Figure 2c and 2d and matched the caption descriptions of Figure 2c and 2d with the contents.

Reviewer 2 Report

  1. The Abstract of the manuscript-at-hand appears quite shallow and needs to be documented in a categorical manner, i.e., it needs to delineate the significance of the research-at-hand succinctly by primarily elaborating on the overall domain, illustrating the underlying challenges of the said domain, and summarizing the key challenges especially addressed by this particular manuscript. The same is also true for the Introduction section.
  2. The contributions of the manuscript-at-hand need to be documented at the end of the Introduction section preferably in a serialized manner, i.e., as (1)__; (2)__; and (3)__.
  3. The manuscript-at-hand also does not have the indispensable section of Literature Review, i.e., the authors have not presented an overview of the state-of-the-art primarily in terms of pros and cons so as to justify both the need and the novelty of the undertaken research work.
  4. The referred work in the Bibliography is also pretty outdated, i.e., most of the referred papers are from the years, 2014 - 2016, with one each from 2017 and 2018, and only one from 2020.
  5. The section, Experiment, also needs to be considerably extended. Also, it needs to be delineated in a more illustrative manner so that there lies no ambiguity in the same for a novice reader.
  6. Critical analysis of the Results is also missing here, i.e., a mere illustration of the same has been presented and which is already evident from the Figures themselves.
  7. There are a considerable number of issues with the grammar, jargon, and sentence structure and a careful proofreading is indispensable here, e.g., lines 144 - 145 state, "To evaluated effectiveness of this device structure optimization on the long-term accuracy of the trained network We implement ...", and lines 169 - 172 state, "In order to develop the long-term accuracy and reduce the refresh rate in trained BNN, an optimized 3D VRRAM with the structure of TiN/TiOx/HfOx/Ti/Pd to implement the BNN", and so on.

Author Response

Revisions to Manuscript ID micromachines-1581488

We appreciate your efforts in the timely review of our paper. We'd like to resubmit our revision of Manuscript ID micromachines-1581488, entitled " Enhancement of long-term accuracy for binary neural networks based on optimized three-dimensional memristor array”, including revisions to the concerns brought up by the reviewers. The editor’s and reviewers’ suggestions were very helpful and served to improve the manuscript. According to their suggestions and feedback, we have completed a major revision to our manuscript. The detailed answers to all the comments are summarized as follows:

Comments from Reviewer 2

Comment 1: The Abstract of the manuscript-at-hand appears quite shallow and needs to be documented in a categorical manner, i.e., it needs to delineate the significance of the research-at-hand succinctly by primarily elaborating on the overall domain, illustrating the underlying challenges of the said domain, and summarizing the key challenges especially addressed by this particular manuscript. The same is also true for the Introduction section.

Reply to Comment 1: Thank you very much for your valuable comment. We reorganized the abstract and Introduction sections. In the revised manuscript, the In embedded neuromorphic Internet of Things (IoT) systems, it is critical to improve the efficiency of neural network (NN) edge devices in inferring a pretrained NN. And device integration, data retention characteristics and power consumption are particularly important in edge computing. In this paper, the self-selected device (SSD), which is the base cell for building the densest three-dimensional (3D) architecture, is used to store non-volatile weights in the binary neural networks (BNN) for embedded NN applications.

Comment 2: The contributions of the manuscript-at-hand need to be documented at the end of the Introduction section preferably in a serialized manner, i.e., as (1)__; (2)__; and (3)__.

Reply to Comment 2: Thank you very much for your valuable comment. In this work, firstly, we fabricated the SSDs with the structure of TiN/TiOx/HfOx /Ru and TiN/TiOx/HfOx/Ti/Ru are tested the electrical propertied including operation voltages, nonlinearity and retention characteristics. We proposed that the underling mechanisms for LCS (low conductance states) and HCS (high conductance states) are due to Poole-Frenkel (PF) emission and trap-assisted-tunneling (TAT). Then, we give the mecha-nism of retention degradation through a series of comparative experiments. By introduc-ing a deep level trap in this structure with Ti interfacial layer, the oxygen ions are firmly trapped and the retention is highly developed. Thirdly, we simulated the long-term accu-racy of the BNN through fashion-MNIST tasks by mapping binarized weights to the non-volatile 3D memristor arrays constructed by these SSDs. Due to the improvement of device retention characteristics, the trained network can guarantee good recognition ac-curacy in 105 seconds, which is estimated to be up to 3 years. At the same time, this scheme can reduce the training energy consumption of the network by 25 thousand times since the network needs to be retrained when the recognition accuracy is lower than 80%. This optimized 3D vertical RRAM provides a choice to provide high-density storage and non-volatile means to meet the requirements of embedded neuromorphic applications for low power consumption and miniaturization.

Comment 3: The manuscript-at-hand also does not have the indispensable section of Literature Review, i.e., the authors have not presented an overview of the state-of-the-art primarily in terms of pros and cons so as to justify both the need and the novelty of the undertaken research work.

Reply to Comment 3: Thank you very much for your valuable comment.

In the Introduction section of the revised edition, we added the following notes:“Currently, the resource consumption of add-ons has become a limitation in memris-tor-based analog computing in memory systems. The analog designs require additional circuits, such as analog-to-digital converters and digital-to-analog converters, to fight against undesirable device properties. In contrast, binary network has obvious advantages in speed, energy consumption, memory occupation and other aspects. However, more storage units are required as weights storage for BNN. Although several software algo-rithm solutions, such as sparse mapping schemes, have been proposed to address the large number of weights [8-9]. The neuromorphic architectures still demand a high amount of storage. 3D memristor array would be the most effective hardware scheme that can maximize the area efficiency [10-12]. However, the retention of written data on the de-vice could affect the energy efficiency of the system operation [11-12].”

[8]   Corey Lammie, Jason K. Eshraghian, Chenqi Li, Amirali Amirsoleimani, Roman Genov, Wei D. Lu, Mostafa Rahimi Azghadi., "Design Space Exploration of Dense and Sparse Mapping Schemes for RRAM Architectures." arXiv preprint, 2022.

[9]   Matthieu Courbariaux, Itay Hubara, Daniel Soudry, Ran El-Yaniv, Yoshua Bengio., "Binarized Neural Networks: Training Deep Neural Networks with Weights and Activations Constrained to +1 or -1." arXiv preprint, 2016.

[10]  B. Kim, E. Hanson and H. Li., "An Efficient 3D ReRAM Convolution Processor Design for Binarized Weight Networks," in IEEE Transactions on Circuits and Systems II: Express Briefs, 2021, 68, 1600-1604.

Comment 4: The referred work in the Bibliography is also pretty outdated, i.e., most of the referred papers are from the years, 2014 - 2016, with one each from 2017 and 2018, and only one from 2020.

Reply to Comment 4: Thank you very much for your valuable comment. We have updated the latest literature in the revsed manuscrip.

Comment 5: The section, Experiment, also needs to be considerably extended. Also, it needs to be delineated in a more illustrative manner so that there lies no ambiguity in the same for a novice reader.

Reply to Comment 5: Thank you very much for your valuable comment. In the revision, we have extended the detail process flow in the Experiment section. “Firstly, Multiple TiN (60nm)/SiO2 (100nm) layers were subsequently deposited by PVD and PECVD. Then, the trenches with smooth sidewall profiles were formed one-step etch-ing. TiN layers, serving as the bottom electrode of the self-selective cell, denoted as the word lines WLs in the 3D memory array. The selective layers prepared by plasma oxida-tion process were self-aligned with the bottom electrode, and its thicknesses is controlled by the oxidation time. After depositing the HfO2 switching layers with different thickness-es on the sidewalls by atomic layer deposition (ALD), the top electrode W (60 nm) with a line width (1μm) was sequentially deposited by magnetron sputtering. Finally, stair-like WLs were opened by selective etching in sequence, and bit-lines (BL) were formed by a lift-off process. The area of the memory cell was defined by the thickness of the bottom electrode TiN and the width of the lateral BL, i.e., 0.06um2.”

Comment 6: Critical analysis of the Results is also missing here, i.e., a mere illustration of the same has been presented and which is already evident from the Figures themselves.

Reply to Comment 6: Thank you very much for your valuable comment. We have extende critical analysis in the Results and Discussion sections for the revison. In page 3, lines 127 - 131 state: The on/off ratios collected from 20 different devices were shown in Figure 1e. It can be seen that the on/off ratio for the DWT is 1 order higher than that of DOT, due primarily to the increased resistance of the low resistance state, which were consistent with the I-V char-acteristics in Figure 1d. In page 4, lines 132 - 135 state: As shown in figure 1e, the NR for DOT is calculated from the currents of 2 V and 1 V, while NR of DWT is calculated at at 1.5 V and 0.75 V. The nonlinearity of DWT device is smaller than that of DOT, and therefore, the integration scale will be sacrificed when this SSD is integrated into a 3D architecture. In page 5, lines 178 - 183 state: During the retention test for DWT, titanium interfacial layer retains oxygen ions and pre-vents the diffusion of oxygen ions toward to the switching layer. In this case, the recombi-nation probability of Vo and oxygen ions is reduced, effectively improving the retention characteristics of the HCS of the SSD. In addition, titanium is a CMOS-compatible material and can be widely used in mass production. In page 5, lines 191 - 192 state: “indicating that the introduction of the Ti layer greatly improves the data retention proper-ties.”.

Comment 7: There are a considerable number of issues with the grammar, jargon, and sentence structure and a careful proofreading is indispensable here, e.g., lines 144 - 145 state, "To evaluated effectiveness of this device structure optimization on the long-term accuracy of the trained network We implement ...", and lines 169 - 172 state, "In order to develop the long-term accuracy and reduce the refresh rate in trained BNN, an optimized 3D VRRAM with the structure of TiN/TiOx/HfOx/Ti/Pd to implement the”

Reply to Comment 7: Thank you very much for your valuable comment.

We have proofreaded the manuscript with the help of professionals. Details are as follows:lines 144 - 145 state, "To evaluated effectiveness of this device structure optimization on the long-term accuracy of the trained network We implement ...", has modified as “To evaluate the effectiveness of this device structure optimization on the long-term accuracy of the trained network, a multi-layer perceptron BNN is implemented for the Fashion-MNIST classification task using a 3D VRRAM array, as shown in Figure 2b.” and lines 198 - 200 state, "In order to develop the long-term accuracy and reduce the refresh rate in trained BNN, an optimized 3D VRRAM with the structure of TiN/TiOx/HfOx/Ti/Ru to implement the…….” has modified as “In order to improve the long-term accuracy and reduce the refresh rate of BNN after train-ing, an optimized 3D VRRAM with the structure of TiN/TiOx/HfOx/Ti/Ru is proposed to implement BNN.”

Reviewer 3 Report

This article is very interesting, but some minor modifications should be noted. Please refer to the following comments:

1. There is a 6 order of magnitude difference between the HRS and LRS conductance presented in Fig 1d, 2c and 2d, with those reported in Fig. 3. This deserves at least an explanation.

2. The standard for referring to the different conductive states of an RRAM device is HRS (high resistance state) and LRS (low resistance state), why used LCS / HCS to show in Fig 3.

3. What is the size of the structure being proved in figure 1 and two? That is, what is the height and width of the VRRAM cell? (which would be the equivalent to the lateral size of a standard planar RRAM cell).

4. There are some grammatical and semantical errors in the text. The authors should carefully look at the overall manuscript to correct the grammatical and semantical errors in the text. In addition, Figures 2c and 2d are wrong with the caption description.

Author Response

Revisions to Manuscript ID micromachines-1581488

We appreciate your efforts in the timely review of our paper. We'd like to resubmit our revision of Manuscript ID micromachines-1581488, entitled " Enhancement of long-term accuracy for binary neural networks based on optimized three-dimensional memristor array”, including revisions to the concerns brought up by the reviewers. The editor’s and reviewers’ suggestions were very helpful and served to improve the manuscript. According to their suggestions and feedback, we have completed a major revision to our manuscript. The detailed answers to all the comments are summarized as follows:

Comments from Reviewer 3

This article is very interesting, but some minor modifications should be noted. Please refer to the following comments:

Comment 1: There is a 6 order of magnitude difference between the HRS and LRS conductance presented in Fig 1d, 2c and 2d, with those reported in Fig. 3. This deserves at least an explanation.

Reply to Comment 1: Thank you very much for your valuable comment. For fig 1d, 2c and 2d, the unit of current on the Y-axis is amperes. However, the unit of conductance is microSiemens in Fig.3. So, there is 6 order of magnitude difference.

Comment 2: The standard for referring to the different conductive states of an RRAM device is HRS (high resistance state) and LRS (low resistance state), why used LCS / HCS to show in Fig 3.

Reply to Comment 2: Thank you very much for your valuable comment. The RRAM devices are wildly studied as the basic unit for memory and computing. In memory applications, the conductive state of RRAM devices is represented by HRS and LRS.  In computational applications, conductive states can also be represented by HCS and LCS. Here, we use LCS/HCS in the article. There is a quantitative relationship between resistance and conductance, i.e. I = VxG.

Comment 3: What is the size of the structure being proved in figure 1 and two? That is, what is the height and width of the VRRAM cell? (Which would be the equivalent to the lateral size of a standard planar RRAM cell).

Reply to Comment 3: Thank you very much for your valuable comment. The size of the RRAM cell is equal to the product of thickness of the stack and the width of Bitline. In this paper, the thickness of the stack is 60nm and the width of Bitline is 1um. Therefore, the size of the RRAM cell is 60nm*1um =0.06um2. In the revision, we have added this information in the Experiment section. Firstly, Multiple TiN (60nm)/SiO2 (100nm) layers were subsequently deposited by PVD and PECVD. Then, the trenches with smooth sidewall profiles were formed one-step etch-ing. TiN layers, serving as the bottom electrode of the self-selective cell, denoted as the word lines WLs in the 3D memory array. The selective layers prepared by plasma oxida-tion process were self-aligned with the bottom electrode, and its thicknesses is controlled by the oxidation time. After depositing the HfO2 switching layers with different thicknesses on the sidewalls by atomic layer deposition (ALD), the top electrode W (60 nm) with a line width (1μm) was sequentially deposited by magnetron sputtering. Finally, stair-like WLs were opened by selective etching in sequence, and bit-lines (BL) were formed by a lift-off process. The area of the memory cell was defined by the thickness of the bottom electrode TiN and the width of the lateral BL, i.e., 0.06um2.

Comment 4: There are some grammatical and semantical errors in the text. The authors should carefully look at the overall manuscript to correct the grammatical and semantical errors in the text. In addition, Figures 2c and 2d are wrong with the caption description.

Reply to Comment 4: Thank you very much for your kindly reminder.(1)   We have carefully revised the manuscript and corrected the grammatical and semantical errors in the text.  (2) We have swapped the positions of Figure 2c and 2d and matched the caption descriptions of Figure 2c and 2d with the contents. 

Round 2

Reviewer 2 Report

Thank you for addressing the comments to a somewhat reasonable extent. However, it would be appreciated if the authors delineate on the notion of 'energy-efficient artificial intelligence technology for IoT' in a bit of a more comprehensive manner duly supported by references. The authors are thus advised to refer to the following literature in this regard:

  • W. E. Zhang et al., "The 10 Research Topics in the Internet of Things", 2020 IEEE 6th International Conference on Collaboration and Internet Computing (CIC), 2020, pp. 34-43.
  • S. Zhu, K. Ota, and M. Dong, "Energy Efficient Artificial Intelligence of Things with Intelligent Edge," in IEEE Internet of Things Journal.

The illustration pertinent to the Literature Review should be elaborated too (the pros and cons of each of the referred studies should be delineated in a categorical manner). Also, mere three references would not do any justice to the existing state-of-the-art, and therefore, the authors should refer to some more relevant literature in this regard.

The language of the manuscript-at-hand needs to be carefully revisited again. There are a considerable number of issues pertinent to the sentence structure, jargon, and punctuation. Please also see the following concerns:

  • Line 13: And device integration, data retention characteristics and power consumption are particularly important in edge computing (a sentence cannot start with 'And').
  • Lines 128 - 131: It can be seen that the on/off ratio for the DWT is 1 order higher than that of DOT, due primarily to the increased resistance of the low resistance state, which were consistent with the I-V characteristics in Figure 1d ('due primarily to' should be written as 'primarily due to').
  • Lines 132 - 133: As shown in figure 1e, the NR for DOT is calculated from the currents of 2 V and 1 V, while NR of DWT is calculated at at 1.5 V and 0.75 V (at at?)

Author Response

Revisions to Manuscript ID micromachines-1581488

We appreciate your efforts in the timely review of our paper. We'd like to resubmit our revision of Manuscript ID micromachines-1581488, entitled " Enhancement of long-term accuracy for binary neural networks based on optimized three-dimensional memristor array”, including revisions to the concerns brought up by the reviewers. The editor’s and reviewers’ suggestions were very helpful and served to improve the manuscript. According to their suggestions and feedback, we have completed a major revision to our manuscript. The detailed answers to all the comments are summarized as follows:

Comments from Reviewer 2

Comment 1: Thank you for addressing the comments to a somewhat reasonable extent. However, it would be appreciated if the authors delineate on the notion of 'energy-efficient artificial intelligence technology for IoT' in a bit of a more comprehensive manner duly supported by references. The authors are thus advised to refer to the following literature in this regard:

  1. E. Zhang et al., "The 10 Research Topics in the Internet of Things", 2020 IEEE 6th International Conference on Collaboration and Internet Computing (CIC), 2020, pp. 34-43.
  2. Zhu, K. Ota, and M. Dong, "Energy Efficient Artificial Intelligence of Things with Intelligent Edge," in IEEE Internet of Things Journal.

The illustration pertinent to the Literature Review should be elaborated too (the pros and cons of each of the referred studies should be delineated in a categorical manner). Also, mere three references would not do any justice to the existing state-of-the-art, and therefore, the authors should refer to some more relevant literature in this regard.

Reply to Comment 1: Thank you very much for your valuable suggestion. We reorganized the Introduction sections. Energy is a crucial resource for smart devices in the Internet of Things (IoT), as most of the applications are powered by battery or uses energy harvesting techniques [1-4]. Be-cause of this, energy-efficient artificial intelligence technologies are becoming increasingly important for the IoT.

[1] Zhu, Sha, Kaoru Ota, and Mianxiong Dong. "Energy Efficient Artificial Intelligence of Things with Intelligent Edge." IEEE Internet of Things Journal, 2022.

[2] Zhang, Wei Emma, et al. "The 10 research topics in the Internet of Things." 2020 IEEE 6th International Conference on Collaboration and Internet Computing (CIC). IEEE, 2020.

[3] L. Farhan, R. Kharel, O. Kaiwartya, M. Quiroz-Castellanos, A. Alissa and M. Abdulsalam, "A Concise Review on Internet of Things (IoT) -Problems, Challenges and Opportunities," 2018 11th International Symposium on Communication Systems, Networks & Digital Signal Processing (CSNDSP), 2018, pp. 1-6

[4] Khayyam, Hamid, et al. "Artificial intelligence and internet of things for autonomous vehicles." Nonlinear approaches in engineering applications. Springer, Cham, 2020. 39-68.

Comment 2: The Line 13: And device integration, data retention characteristics and power consumption are particularly important in edge computing (a sentence cannot start with 'And').

Lines 128 - 131: It can be seen that the on/off ratio for the DWT is 1 order higher than that of DOT, due primarily to the increased resistance of the low resistance state, which were consistent with the I-V characteristics in Figure 1d ('due primarily to' should be written as 'primarily due to').

Lines 132 - 133: As shown in figure 1e, the NR for DOT is calculated from the currents of 2 V and 1 V, while NR of DWT is calculated at at 1.5 V and 0.75 V (at at?)

Reply to Comment 2: Thank you very much for your valuable comment. We have proofreaded the manuscript with the help of professionals. Details are as follows: lines 13-14 state, “And device integration, data retention characteristics and power consumption are particularly important in edge computing”, has modified as “ Meanwhile, in edge computing, device integration, data retention characteristics and power consumption are particularly important.” Lines 131-134: “It can be seen that the on/off ratio for the DWT is 1 order higher than that of DOT, due primarily to the increased resistance of the low resistance state, which were consistent with the I-V characteristics in Figure 1d.”, has modified as “It can be seen that the on/off ratio for the DWT is 1 order higher than that of DOT, primarily due to the increased resistance of the low resistance state, which is consistent with the I-V characteristics in Figure 1d.” Lines 135 - 136: “As shown in figure 1e, the NR for DOT is calculated from the currents of 2 V and 1 V, while NR of DWT is calculated at at 1.5 V and 0.75 V”, has modified as “As shown in figure 1e, the NR for DOT is calculated from the currents at 2 V and 1 V, while that for DWT, at 1.5 V and 0.75 V”

Reviewer 3 Report

Accept in present form

Author Response

Thank you.

Round 3

Reviewer 2 Report

Thank you for addressing the comments. The quality of the manuscript-at-hand has accordingly improved.